# The Association between Influenza and Pneumococcal Vaccinations and SARS-Cov-2 Infection: Data from the EPICOVID19 Web-Based Survey

**DOI:** 10.3390/vaccines8030471

**Published:** 2020-08-23

**Authors:** Marianna Noale, Caterina Trevisan, Stefania Maggi, Raffaele Antonelli Incalzi, Claudio Pedone, Mauro Di Bari, Fulvio Adorni, Nithiya Jesuthasan, Aleksandra Sojic, Massimo Galli, Andrea Giacomelli, Sabrina Molinaro, Fabrizio Bianchi, Claudio Mastroianni, Federica Prinelli, on behalf of the EPICOVID19 Working Group

**Affiliations:** 1National Research Council-Neuroscience Institute, Aging Branch, Via Vincenzo Maria Gallucci 16, 35128 Padova, Italy; caterina.trevisan.5@studenti.unipd.it (C.T.); stefania.maggi@in.cnr.it (S.M.); 2Geriatric Unit, Department of Medicine (DIMED), University of Padova, Via Giustiniani 2, 35128 Padova, Italy; 3Unit of Geriatrics, Department of Medicine, Biomedical Campus of Rome, via Alvaro del Portillo, 21, 00128 Rome, Italy; r.antonelli@unicampus.it (R.A.I.); claudio.pedone@gmail.com (C.P.); 4Geriatric Intensive Care Medicine, University of Florence and Azienda Ospedaliero-Universitaria Careggi, Viale Peraccini 18, 50139 Florence, Italy; mauro.dibari@unifi.it; 5National Research Council-Institute of Biomedical Technologies, Epidemiology Unit, Via Fratelli Cervi 93, 20090 Segrate, Italy; fulvio.adorni@itb.cnr.it (F.A.); nithiya.jesuthasan@itb.cnr.it (N.J.); aleksandra.sojic@itb.cnr.it (A.S.); federica.prinelli@itb.cnr.it (F.P.); 6Infectious Diseases Unit, Department of Biomedical and Clinical Sciences L. Sacco, Università di Milano, ASST Fatebenefratelli Sacco, 20157 Milan, Italy; massimo.galli@unimi.it (M.G.); andrea.giacomelli@unimi.it (A.G.); 7National Research Council-Institute of Clinical Physiology, Epidemiology and Health Research Laboratory, Via G. Moruzzi 1, 56124 Pisa, Italy; sabrina.molinaro@ifc.cnr.it; 8National Research Council-Institute of Clinical Physiology, Department of Environmental Epidemiology and Disease registries, Via G. Moruzzi 1, 56124 Pisa, Italy; fabriepi@ifc.cnr.it; 9Public Health and Infectious Disease Department, “Sapienza” University, Piazzale Aldo Moro 1, 00185 Rome, Italy; claudio.mastroianni@uniroma1.it

**Keywords:** SARS-CoV-2, COVID-19, web-based survey, nasopharyngeal swab testing, influenza vaccination, pneumococcal vaccination

## Abstract

The present study aims to evaluate whether influenza and pneumococcal vaccinations are associated with positive nasopharyngeal swab (NPS) testing to detect SARS-CoV-2. Data from the Italian cross-sectional web-based survey (EPICOVID19), based on a self-selection sample of individuals aged ≥18, were considered. The probability of a positive SARS-CoV-2 NPS test result as a function of influenza or anti-pneumococcal vaccination was evaluated using multivariable logistic regression, stratifying analysis by age (<65 years, ≥65 years). From April 2020, 170,731 individuals aged <65 years and 28,097 ≥65 years filled out the EPICOVID19 questionnaire. Influenza and anti-pneumococcal vaccinations were received, respectively, by 16% and 2% of those <65 years, and by 53% and 13% of those ≥65 years. SARS-CoV-2 NPS testing was reported by 6680 participants. Anti-pneumococcal and influenza vaccinations were associated with a decreased probability of a SARS-CoV-2 NPS positive test in the younger participants (OR = 0.61, 95% CI 0.41–0.91; OR = 0.85, 95%CI 0.74–0.98; respectively). A significantly lower probability of a positive test result was detected in the individuals ≥65 years who received anti-pneumococcal vaccination (OR = 0.56, 95%CI 0.33–0.95). These results need to be confirmed by further investigations, but they are relevant given the probable coexistence of influenza, bacterial infections, and COVID-19 over the coming autumn–winter season.

## 1. Introduction

The COronaVIrus Disease 19 (COVID-19) outbreak continues to be a burdensome, multifaceted public health concern, and as of July 17, 2020, the number of confirmed worldwide cases has risen to 13,788,300 (with America, Asia and Europe the most affected continents); it has also been responsible for 589,688 deaths [1]. The disease burden of the pandemic in terms of the percentage of the population affected and the severity of symptoms has been high especially for older individuals. In Western countries [2] in particular, senior adults have accounted for almost half of the total confirmed cases [3,4] and more than two thirds of the deaths due to the infection [5]. The lack of specific therapies for COVID-19 has prompted a growing number of clinicians and researchers to investigate risk factors that may affect individual susceptibility to the infection and disease severity.

A factor contributing to the higher vulnerability of older adults to SARS-CoV-2 could be linked to aging-related immunologic changes [6,7]. These include both immunosenescence, i.e., the progressive dysfunction of both the innate and adaptive immune responses, and the development of low-grade chronic systemic inflammation, termed inflammaging [7,8]. In addition to increasing susceptibility and altering the response to infectious diseases, these changes appear to slightly reduce the efficacy of vaccines in older individuals [9,10,11,12,13]. Nonetheless, the clinical effectiveness of vaccines in older people is supported not only by vaccine-specific responses, but also by the cross-reactivity, cross-protection and immunostimulation linked to the vaccines per se and to their adjuvants [8,14]. Vaccines are thus able to protect individuals against the targeted pathogen as well as to reduce the associated risk of viral or bacterial co-infections [15].

These considerations have led to the hypothesis that influenza and pneumococcal vaccinations could stimulate an immune response in older adults and potentially lower the risk and the severity of other infections, including COVID-19. Despite the current scarcity of data supporting this hypothesis [16,17,18,19], it is a particularly timely issue since influenza and pneumococcal vaccines are easily accessible at a time that public health officials are facing the challenge of the 2020 autumn–winter season without an effective vaccine for COVID-19 [20,21].

The data presented here were collected by the EPICOVID19, a self-administered, volunteer, web-based survey investigating the number of suspected cases of COVID-19 and the potential determinants of SARS-CoV-2 infection in a large sample of Italian respondents [22]. The current study aimed to evaluate if influenza and pneumococcal vaccinations are associated with a lower probability of positivity to nasopharyngeal swab (NPS) detection of SARS-CoV-2 in adults and older individuals.

## 2. Materials and Methods 

### 2.1. Study Design and Setting

EPICOVID19 is an Italian cross-sectional, web-based survey that was initiated in April 2020. A national convenience sample of volunteers was considered. The participants were recruited via social media (Facebook, Twitter, Instagram, Whatsapp), press releases, internet pages, local radio, TV stations, and institutional websites; the European Commission's open-source official EUSurvey management tool was used to collect the data (https://epicovid19.itb.cnr.it/). The inclusion criteria for participation in the survey were: (i) being 18 or older; (ii) having access to a mobile phone, computer, or tablet with internet connectivity; (iii) giving informed on-line consent to participate in the study. Those who did not meet these requirements were excluded from the study sample. The study was registered (ClinicalTrials.gov NCT04471701).

### 2.2. Data Collection and Variables

The self-administered questionnaire included 38 mainly mandatory, closed questions, divided into six sections: socio-demographic data; clinical evaluation; personal characteristics; housing conditions; lifestyle; behaviors following the lockdown. The socio-demographic variables considered for the purposes of this study included sex, age and educational level (primary school or less, middle or high school, and university degree or post-graduate degree). The regions of residence were grouped into four areas according to their ratio of total swabs performed/total individuals tested at least once and total COVID-19 cases/total individuals tested at least once [23]; a higher ratio corresponded to more regional resources allocated for testing. As illustrated in Appendix A, the four areas identified in ascending order were: 1. Piedmont, Lombardy, Aosta Valley, Emilia Romagna, Liguria and Marche; 2. Tuscany, Trentino Alto Adige, Abruzzo, Apulia; 3. Veneto, Friuli Venezia Giulia, Lazio, Molise, Campania; 4. Sicily, Sardinia, Umbria, Calabria and Basilicata.

The clinical evaluation section took into consideration symptoms reported from February (fever >37.5° for at least three consecutive days, sore throat/rhinorrhea, cough, headache, myalgia, anosmia/dysgeusia, shortness of breath, chest pain, heart palpitations, gastrointestinal disturbances, conjunctivitis; pneumonia). Self-reported chronic conditions and regularly taken medications allowed us to derive the presence of lung diseases, cardiovascular diseases (CVD), hypertension, oncological diseases, liver diseases, renal diseases, immune system diseases, metabolic diseases, thyroid diseases, depression and/or anxiety.

The respondent was also asked if he/she had received an influenza vaccination during the autumn of 2019 and/or an anti-pneumococcal vaccination over the last 12 months, if he/she had undergone NPS testing for SARS-CoV-2 and its result (positive vs. negative), and if he/she had contacts with confirmed COVID-19 cases. Questions about autonomy in carrying out daily activities, self-rated health (very bad, bad, adequate, good or very good) and smoking status (never, former and current smoker) were also available.

### 2.3. Ethics and Consent Form

The Ethics Committee of the National Institute for Infectious Diseases (Italian: Istituto Nazionale per le Malattie Infettive I.R.C.C.S. Lazzaro Spallanzani) approved the EPICOVID19 study protocol (Protocol No. 70, 12/4/2020). The participants were requested to give their informed consent when they first accessed the on-line platform. The study was carried out in accordance with the principles of the Declaration of Helsinki.

All data were handled and stored in accordance with the European Union General Data Protection Regulation (EU GDPR) 2016/679 [24]; data transfer included encrypting/decrypting and password protection.

### 2.4. Statistical Analysis

The categorical variables were presented as counts and percentages; the continuous variables were summarized using means and standard deviations (SD). The comparison between the participants’ characteristics classified in accordance to (a) having had an influenza vaccination during autumn of 2019, (b) having had an anti-pneumococcal vaccination during the last 12 months, and (c) the results of a NPS test (positive vs. negative) were assessed using the Chi-squared test for categorical variables. Generalized linear models were used for the continuous variables after testing for homoschedasticity (Levene test).

Crude and adjusted logistic regression models were built to test the probability of a positive NPS SARS-CoV-2 test as a function of having had the influenza or pneumococcal vaccinations. Models were stratified by age classes (<65, ≥65 years) and were adjusted for age, sex, education, area of residence, self-reported comorbidities (CVD, hypertension, lung diseases, diabetes treated with medications, metabolic diseases) and smoking status. The linearity assumption for quantitative variable (age) was evaluated considering an analysis of quartiles [25]. Odds ratios (ORs) were presented with their 95% confidence intervals (CIs). Two-tail p-values <0.05 were considered statistically significant. The analyses were performed using SAS statistical package, version 9.4 (SAS Institute Inc., Cary, NC, USA).

## 3. Results

The data were collected between April and June 2020 from questionnaires completed by 198,828 respondents with a mean (SD) age of 48 (14.7) years (59.7% female). A total of 28,097 participants were aged between 65–104 years (mean (SD) age 70.8 (5.5) years, 50.9% females) and 170,731 between 18–64 years (mean (SD) age 44.2 (12.1) years, 61.1% females). The participants’ general characteristics are outlined in Table 1. The rates of response for each Italian Region x 10,000 inhabitants are shown in Appendix A.

The distributions of age, sex and education were scarcely representative of the general Italian population, as outlined by the Italian National Institute of Statistics (ISTAT) in their figures for 2019 (Appendix A) [26,27]: of note, the survey respondents had a higher level of education (more than 50% had University or post graduate degree vs. less than 10% according to ISTAT data).

Fifteen thousand and eleven respondents 65 or older (53.4%) had received influenza vaccination during the 2019 autumn; 3461 (12.3%) had received anti-pneumococcal vaccination. The respective percentages in the younger group were 15.7% and 2.2%. With respect to their unvaccinated counterparts, the participants who received the influenza vaccination were more likely to be older, male, to have higher levels of education, and a higher prevalence of hypertension, metabolic diseases, CVD, cancer, lung diseases, and diabetes treated with medications (Appendix A). Similar differences in educational level and chronic disease prevalence were observed comparing the respondents who received anti-pneumococcal vaccination and their unvaccinated counterparts (Appendix A).

Six thousand six hundred and eighty participants underwent a SARS-CoV-2 NPS test (25.1% positive vs. 74.9% negative results, Table 2). The participants 65 and over with a positive test result (n = 320) were significantly older (76.9 (SD = 9.7) vs. 71.4 (SD = 7.7) years), more frequently women (57.2% vs. 41.5%) had lower levels of education with respect to those with a negative result. Older respondents with positive test result had higher prevalence of CVD and dependency in daily activities, and a lower percentage of current smokers than among those with negative results. Participants younger than 65 with a positive test result (n = 1356) were found to be significantly older (46.5 (SD = 11.5) years vs. 44.8 (SD = 11.4)), less frequently women (61.9% vs. 69.2%), had lower levels of education, and had higher prevalence of metabolic diseases than those with a negative test result. Older respondents with a positive test result reported a significantly lower percentage of anti-pneumococcal vaccination over the last 12 months (10% vs. 21.1%). Similar results were observed comparing the younger participants with a positive vs. negative NPS test for the frequency of both antipneumococcal (2.6% vs. 4.1%) and influenza vaccination (28.2% vs. 31.9%).

At multivariable logistic regression (Table 3), the anti-pneumococcal vaccination in the older participants was significantly associated with a decreased odds of positive SARS-CoV-2 test (OR = 0.56, 95%CI: 0.33–0.95). Analyses on those aged <65 years confirmed a significantly lower probability of a positive test following anti-pneumococcal vaccination (OR = 0.61, 95%CI: 0.41–0.91) and a borderline lower probability for the influenza vaccination (OR = 0.85, 95%CI: 0.74–0.98).

## 4. Discussion

The study found that the adult and older EPICOVID19 respondents who had received an anti-pneumococcal vaccination in the previous year had a lower probability of having a positive SARS-CoV-2 test with respect to their non-vaccinated counterparts. Considering influenza vaccination, only the participants aged <65 years had a lower probability of being infected with SARS-CoV-2 infection with respect to their non-vaccinated ones.

An analysis of our data showed that the older cohort (65 and older) of the online survey sample was poorly representative of the older Italian population both in terms of age and educational level [26,27]. This may have been due to the web-based nature of the survey, requiring a certain degree of technological skills to participate. Indeed, with respect to the general Italian older population, the older EPICOVID19 participants included higher percentages of individuals falling into younger old age classes and with a higher educational level. Despite the fact that a profile such as this generally mirrors high health literacy and awareness of healthy behaviors, the vaccination rates reported in this group were in line with the national statistics, which are far below the proposed target coverage of 75% for influenza and 50% for pneumococcal vaccines [22] for this age class. In particular, as regards the influenza vaccination, the Italian Ministry of Health and the National Institute of Health reported a coverage of 53.1% for individuals ≥65 [28] for the 2018–2019 season, which in our survey was 53.4%. In regards to anti-pneumococcal vaccination, national data are scarce and show wide variability, ranging from 0.7% to 50% in different Italian regions [29]. This issue and the fact that the survey collected data only on the anti-pneumococcal vaccination in the last 12 months, made it difficult to estimate and compare the coverage of this vaccine in our web-based sample with that of the general population. An analysis of the characteristics of the vaccinated and non-vaccinated individuals uncovered that individuals from both age groups who had influenza or anti-pneumococcal vaccinations had a higher risk profile in terms of existing chronic diseases. This suggests that, despite the national recommendations for influenza and anti-pneumococcal vaccinations, such preventive measures are still generally addressed to the most at risk individuals.

Our study found that anti-pneumococcal vaccination was associated, respectively, in the younger and older participants, with a 39% and 44% lower probability of a positive result on a NPS test. The influenza vaccination was, instead, associated with a less pronounced, lower probability of a positive result on a NPS test (−15%), but only in the younger age group; among older participants no significant association was observed, but analyses were underpowered (study power for influenza vaccination among older participants <0.20). Whereas two recently published studies suggest that influenza vaccination is associated with lower COVID-19 severity and mortality [16,17], our study for the first time investigated the probability of being infected with SARS-CoV-2 as a function of influenza and anti-pneumococcal vaccinations.

Some hypotheses can be formulated to interpret our findings.

First of all, the decision to be vaccinated may characterize individuals with a higher level of health literacy and awareness [30], attitudes that may be linked to a greater compliance with the measures suggested for COVID-19 prevention, i.e., physical distancing, hygiene/disinfection practices, and use of personal protective equipment, leading to a lower probability of becoming infected. This interpretation is also coherent with the observation that, as mentioned above, the individuals who were vaccinated were more educated and had a higher risk profile in terms of comorbidities, which may have been one of the reasons they sought to avoid possible sources of contagion, considering themselves more vulnerable to possible infections. However, despite the worse risk profile and higher education of the vaccinated survey respondents, vaccinated and non-vaccinated participants expressed almost comparable ratings of their health status, with only a statistically significant, but clinically questionable, 1% difference in the frequency of individuals who defined their health status as very bad or bad. Moreover, there were no significant differences in the frequency of contacts with confirmed COVID-19 cases in the vaccinated and unvaccinated older participants, who denied such contacts in more than 90% of cases. This suggests that awareness and compliance with the widely publicized preventive measures against COVID-19, at least in terms of physical distancing, were comparable between vaccinated and non-vaccinated participants. Additionally, the significant association between vaccination and SARS-CoV-2 test results persisted even after adjustment for educational level and for the main comorbidities and risk behaviors that could influence the probability of individuals getting the infection.

A second issue to be considered in interpreting our results is the potential impact of different regional health policies in regards to both the vaccination coverage and the accessibility to NPS testing. Indeed, if regions with higher coverage of anti-pneumococcal vaccinations were also those with greater health resources to perform NPS tests, and vice versa, we would overestimate the association between vaccination and positive NPS. To test this hypothesis, we compared the frequency distribution of anti-pneumococcal vaccination and of health resources for NPS testing in different Italian regions. Despite not observing a large overlap between these aspects, we found some similar trends, especially in the Northern Italian regions (Appendix A). However, our results were confirmed both when we either adjusted for or stratified by region of residence, suggesting that this point may have influenced, but cannot completely explain, the observed association.

Although the above considerations on behavioral- and geographic-related issues need to be taken into account, our results allow us to cautiously speculate about a possible vaccine-related protective effect against SARS-CoV-2. Indeed, although not specific for that pathogen, cross-reactivity, cross-protection and immunostimulation by the two vaccines may have contributed to diminishing the risk of other bacterial or viral infections [8,14]. In that regard, vaccine adjuvants play an important role in enhancing the innate immunity to a vaccine antigen and in triggering a rapid, strong adaptive response to infectious agents [31]. The use of vaccine adjuvants is recommended, especially considering that some categories of individuals, such as older people, have a suboptimal immune response to vaccines due to immunosenescence and inflammaging [7,31,32]. These phenomena seem to affect the efficacy of influenza [9,10] and antipneumococcal vaccines [33], and vaccine-related antibody production, which seems to be weaker and of shorter duration in older with respect to younger individuals [31,34,35].

Our analysis, in particular, uncovered a stronger inverse association between the SARS-CoV-2 NPS results and the anti-pneumococcal vaccine with respect to the influenza vaccine, the latter demonstrating borderline results only for the younger age group. In this regard, it is possible that the polysaccharide nature of the anti-pneumococcal vaccine, the conjugation of the antigen with the non-toxic variant of diphtheria toxin (CRM197), and the use of an additional adjuvant (aluminum phosphate) [31] may have enhanced the immunogenicity of the vaccine to a greater extent with respect to the influenza vaccination, especially in older people.

Finally, recent studies suggested that influenza vaccination could mitigate the severity of COVID-19, either by modulating the immune response or by reducing the risk of co-infections [16,17]. If confirmed, this issue should be considered in interpreting our results since non-vaccinated individuals may have had a more severe COVID-19 onset and course that made them more likely to undergo a NPS compared with vaccinated people. However, further studies are needed to verify these hypotheses and to explore the differences noted between younger and older adults.

Interestingly, we found a lower frequency of current smokers among individuals with a positive NPS test than among those with negative results. This observation is in line with recently published studies indicating an apparent low risk of SARS-CoV-2 infection among smokers [36]; however, these findings need to be confirmed. The association between smoking habit and the risk of SARS-CoV-2 infection will be investigated in detail in a specific paper.

The limitations characterizing our study refer, first of all, to the poor representativeness of the sample, especially for the oldest age group with respect to the Italian general population. This issue and the fact that only a minority of the total survey respondents had available data on SARS-CoV-2 testing, of course, limited the generalizability of our findings. Nevertheless, in light of the youngish, highly educated survey respondents, the bias more likely resulted in an underestimation rather than an overestimation of the studied associations. Secondly, the study design was observational, which is useful mostly for generating hypotheses. Other more appropriate designs are needed to confirm our results. Third, self-reported data are always associated with a certain risk of recall and misclassification biases, which might have influenced our results with possible over- or underestimation. Fourth, as mentioned above, in addition to the lack of details on the specific type of vaccines’ formulation, the data collected on anti-pneumococcal vaccination referred only to the last 12 months, thus no information was available on vaccination in the previous years. However, this later issue could have more likely led to underestimate the association between the vaccine and positive SARS-CoV-2 testing. On the other hand, the strengths of our work include the novelty of the topic, the large sample size of middle-old and old-old age groups, and the wide data collection on the individual’s vaccination history and COVID-19-related behavior, such as possible related symptoms and SARS-CoV-2 NPS results.

## 5. Conclusions

Public health recommendations for the coming autumn–winter season are critical in the light of possible coexistence of bacterial respiratory infections and influenza and COVID-19 outbreaks [20,21]. The analysis of the data gathered by the EPICOVID19 survey suggests that anti-pneumococcal and, to a lesser extent, influenza vaccinations are associated with lower probability of SARS-CoV-2 infection. These results need to be confirmed by future studies. However, considering that multiple respiratory co-infections can frequently lead, especially in older age, to fatal respiratory failure, our findings support the need for collaborative public health programs to enhance the anti-pneumococcal and influenza vaccination campaign over the coming months. Special attention should be paid, in particular, to the most vulnerable categories of individuals who are at higher risk of a negative COVID-19 prognosis.

## Figures and Tables

**Table 1 vaccines-08-00471-t001:** Characteristics of study participants.

Characteristics of Study Participants	All	<65 Years	≥65 Years
	(n = 198,828)	(n = 170,731)	(n = 28,097)
*Socio-demographic characteristics*			
Sex, males, No. (%)	80167 (40.3)	66382 (38.9)	13785 (49.1)
Age, years, mean ± SD	48.0 ± 14.7	44.2 ± 12.1	70.8 ± 5.5
Education, No. (%)			
Elementary school or less	1300 (0.7)	341 (0.2)	959 (3.0)
Middle or High school	79360 (39.9)	66412 (38.9)	12948 (46.1)
University degree or post-graduate	118168 (59.4)	103978 (60.9)	14190 (50.4)
Italian area of residence *, No. (%)			
Area 1	117176 (59.1)	99905 (58.7)	17271 (61.6)
Area 2	25769 (13.0)	22459 (13.2)	3310 (11.8)
Area 3	43107 (21.7)	37271 (21.9)	5836 (20.8)
Area 4	12350 (6.2)	10705 (6.3)	1645 (5.9)
Smoking status, No. (%)			
Never	114049 (57.4)	100499 (58.9)	13550 (48.2)
Former smoker	47779 (24.0)	36979 (21.7)	10800 (38.4)
Current	37000 (18.6)	33253 (19.5)	3747 (13.3)
*Self-reported diseases ***			
Lung diseases, No. (%)	11456 (5.8)	9377 (5.5)	2079 (7.4)
CVD, No. (%)	15284 (7.7)	7702 (4.5)	7582 (27.0)
Hypertension, No. (%)	33320 (16.8)	19509 (11.4)	13811 (49.2)
Oncological diseases, No. (%)	6719 (3.4)	4189 (2.5)	2530 (9.0)
Depression and/or anxiety, No. (%)	20563 (10.3)	16401 (9.6)	4162 (14.8)
Liver diseases, No. (%)	1487 (0.8)	1011 (0.6)	476 (1.7)
Renal diseases, No. (%)	1689 (0.9)	1133 (0.7)	556 (2.0)
Diabetes treated with medications, No. (%)	4594 (2.3)	2494 (1.5)	2100 (7.5)
Metabolic diseases, No. (%)	20261 (10.2)	11379 (6.7)	8882 (31.6)
Thyroid diseases, No. (%)	14717 (7.4)	11596 (6.8)	3121 (11.1)
Diseases of the immune system, No. (%)	17514 (8.8)	14428 (8.5)	3086 (11.0)
Dependency in daily activities, No. (%)	927 (0.5)	393 (0.2)	534 (1.9)
Self-rated health, No. (%)			
Very bad or Bad	1811 (0.9)	1249 (0.7)	562 (2.0)
Adequate	29745 (15.0)	21768 (12.8)	7977 (28.4)
Good or Very good	167272 (84.1)	147714 (86.5)	19558 (69.6)
Influenza vaccination during last autumn, No. (%)	41820 (21.0)	26809 (15.7)	15011 (53.4)
Anti-pneumococcal vaccination in the last 12 months, No. (%)	7178 (3.6)	3717 (2.2)	3461 (12.3)
Contact with confirmed COVID-19 cases, No. (%)	16025 (8.1)	14744 (8.6)	1281 (4.6)

*Area 1: Piedmont, Lombardy, Emilia Romagna, Liguria, Marche, Aosta Valley. Area 2: Tuscany, Trentino Alto Adige, Abruzzo, Apulia. Area 3: Veneto, Lazio, Friuli Venezia Giulia, Molise, and Campania. Area 4: Sicily, Sardinia, Umbria, Calabria, Basilicata. **: defined considering both self-reported diseases and medications used; SD: Standard Deviation.

**Table 2 vaccines-08-00471-t002:** Characteristics of study participants with an available SARS-CoV-2 NPS test result.

Characteristics of Study Participants	<65 Years			≥65 Years		
	SARS-CoV-2 NPS Test		*p*-Value	SARS-CoV-2 NPS Test		*p*-Value
	Negative Result	Positive Result		Negative Result	Positive Result	
	(n = 4705)	(n = 1356)		(n = 299)	(n = 320)	
*Socio-demographic characteristics*						
Sex, males, No. (%)	1450 (30.8)	530 (39.1)	<0.0001	175 (58.5)	137 (42.8)	<0.0001
Age, years, mean ± SD	44.8 ± 11.4	46.5 ± 11.5	<0.0001	71.4 ± 7.7	76.9 ± 9.7	<0.0001
Italian area of residence *, No. (%)			<0.0001			<0.0001
Area 1	2248 (47.9)	942 (69.6)		172 (57.7)	262 (82.1)	
Area 2	641 (13.7)	135 (10.0)		45 (15.1)	18 (5.6)	
Area 3	1524 (32.5)	247 (18.3)		69 (23.2)	31 (9.7)	
Area 4	279 (6.0)	29 (2.1)		12 (4.0)	8 (2.5)	
Education, No. (%)			<0.0001			<0.0001
Elementary school or less	55 (1.2)	12 (0.9)		17 (5.7)	91 (28.4)	
Middle or High school	1129 (24.0)	450 (33.2)		84 (28.1)	120 (37.5)	
University degree or post-graduate	3521 (74.8)	894 (65.9)		198 (66.2)	109 (34.1)	
Smoking status, No. (%)			<0.0001			<0.0001
Never	2945 (62.6)	896 (66.1)		154 (51.5)	218 (68.1)	
Former smoker	917 (19.5)	321 (23.7)		107 (35.8)	83 (25.9)	
Current	843 (17.9)	139 (10.3)		38 (12.7)	19 (5.9)	
*Self-reported diseases ** and health status*						
Lung diseases, No. (%)	341 (7.3)	104 (7.7)	0.5996	38 (12.7)	48 (15.0)	0.4102
CVD, No. (%)	208 (4.4)	75 (5.5)	0.0878	103 (34.5)	146 (45.6)	0.0046
Hypertension, No. (%)	631 (13.4)	205 (15.1)	0.1083	151 (50.5)	181 (56.6)	0.1308
Oncological diseases, No. (%)	123 (2.6)	32 (2.4)	0.6011	40 (13.4)	32 (10.0)	0.1902
Depression and/or anxiety, No. (%)	461 (9.8)	113 (8.3)	0.1046	67 (22.4)	121 (37.8)	<0.0001
Liver diseases, No. (%)	39 (0.8)	6 (0.4)	0.1442	4 (1.3)	10 (3.1)	0.1782
Renal diseases, No. (%)	33 (0.7)	10 (0.7)	0.8891	15 (5.0)	18 (5.6)	0.7364
Diabetes treated with medications, No. (%)	77 (1.6)	30 (2.2)	0.1560	24 (8.0)	32 (10.0)	0.3924
Metabolic diseases, No. (%)	324 (6.9)	121 (8.9)	0.0113	98 (32.8)	89 (27.8)	0.1790
Thyroid diseases, No. (%)	392 (8.3)	90 (6.6)	0.0422	27 (9.0)	39 (12.2)	0.2034
Diseases of the immune system, No. (%)	479 (10.2)	111 (8.2)	0.0290	25 (8.4)	39 (12.2)	0.1182
Dependency in daily activities, No. (%)	73 (1.6)	16 (1.2)	0.3162	41 (13.7)	90 (28.1)	<0.0001
Self-rated health, No. (%)			0.2937			0.0025
Very bad or bad	54 (1.2)	22 (1.6)		19 (6.4)	22 (6.9)	
Adequate	725 (15.4)	219 (16.2)		102 (34.1)	151 (47.2)	
Good or very good	3926 (83.4)	1115 (82.2)		178 (59.5)	147 (45.9)	
Influenza vaccination during last autumn, No. (%)	1502 (31.9)	382 (28.2)	0.0085	182 (60.9)	180 (56.3)	0.2438
Anti-pneumococcal vaccination in the last 12 months, No. (%)	191 (4.1)	35 (2.6)	0.0114	63 (21.1)	32 (10.0)	0.0001
Contact with confirmed COVID-19 cases, No. (%)	2837 (60.3)	957 (70.6)	<0.0001	113 (37.8)	215 (67.2)	<0.0001

* Area 1: Piedmont, Lombardy, Emilia Romagna, Liguria, Marche, Aosta Valley. Area 2: Tuscany, Trentino Alto Adige, Abruzzo, Apulia. Area 3: Veneto, Lazio, Friuli Venezia Giulia, Molise, and Campania. Area 4: Sicily, Sardinia, Umbria, Calabria, Basilicata. **: defined considering both self-reported diseases and medications used. SD: Standard Deviation.

**Table 3 vaccines-08-00471-t003:** Characteristics of study participants associated with SARS-CoV-2 swab positive results (vs. negative results).

	All			<65 Years			≥65 Years		
	OR	95% CI	*p*-Value	OR	95% CI	*p*-Value	OR	95% CI	*p*-Value
(a) Not adjusted model									
Flu shot during last autumn	1.02	0.91–1.15	0.7387	0.86	0.75*–*0.99	0.0301	0.83	0.60*–*1.14	0.2440
Anti-pneumococcal vaccination	0.77	0.58–1.02	0.0635	0.67	0.46*–*0.97	0.0342	0.42	0.26*–*0.66	0.0002
(b) Adjusted model *									
Flu shot during last autumn	0.89	0.78–1.01	0.1408	0.85	0.74*–*0.98	0.0235	0.87	0.59*–*1.28	0.4826
Anti-pneumococcal vaccination	0.56	0.41–0.75	0.0001	0.61	0.41*–*0.91	0.0156	0.56	0.33*–*0.95	0.0313

* adjusted for sex (males vs. females), age (years, continuous for “all model”; dichotomized into <67 vs. ≥67 for “≥65 years model”; dichotomized into ≥54 vs. <54 for “<65 years model”), education (high school diploma, university degree or post-graduate degree vs. middle school, primary school or less), Italian area of residence (area 1 vs. (3,4); area 2 vs. (3,4)), dichotomized self-reported diseases (CVD, hypertension, lung diseases, diabetes treated with medications, metabolic diseases), smoking status (current smoker vs. never smoker or former smoker) and contact with confirmed COVID-19 cases (yes vs. no).

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
