# Peer review of "The Association between Influenza and Pneumococcal Vaccinations and SARS-Cov-2 Infection: Data from the EPICOVID19 Web-Based Survey"

_vaccines, 2020, doi:10.3390/vaccines8030471_

Round 1
Reviewer 1 Report
General comments. The manuscript contains a lot of important information, making for dense reading. Unfortunately the use of fancy sentences does not help. I wish more authors would follow the old adage that information is best conveyed with “simple declarative sentences”. This would certainly help information be conveyed to all readers especially those in the 90% of the world where English is not peoples first language.
Line 189. Meaning was not clear. Could you replace, “with respect to” with “were among”?
Line 189. The result that people who smoke are less likely to get COVID was surprising, because long term smoking can cause irreversible lung damage. Thus, this is a very important observation that should have been dealt with in the discussion. If the smoking result is to described in another paper, then the authors must say so. If the authors did not elaborate on this observation, because others have made it previously they should say so, and provide references.
Further consideration. The discussion of the association with the pneumococcal and flu vaccines with less covid-19 has been well discussed. There is an additional explanation that the authors may wish to include. If co-infections with flu or pneumococci result is a worse course of Covid-19 infection, then such individuals would be more likely to be tested and declared Covid-19 positive. In other words, by keeping flu and bacterial causes of pneumonia under control, the outcomes of co-infections withCovid-19 may be less severe on average.
Author Response
20 August 2020
Subject: Manuscript ID: vaccines-908602 - Minor Revisions (by 24th August 2020)
Dear Editor,
herewith we resubmit the revised version of our manuscript entitled “The association between influenza and pneumococcal vaccinations and SARS-CoV-2 infection: data from the EPICOVID19 web-based survey”.
We are grateful to the reviewers for their time and constructive comments. We have implemented their comments and suggestions and wish to submit a revised version of the manuscript for further consideration in the journal. Below, we provide a point-by-point response to each remark with a description and referral to all amendments made in the revised version of the manuscript. Changes in the manuscript are highlighted using track changes method.
We thank you and look forward to hearing from you.
Yours sincerely,
On behalf of the co-authors
Marianna Noale
Reviewer 1:
General comments. The manuscript contains a lot of important information, making for dense reading. Unfortunately the use of fancy sentences does not help. I wish more authors would follow the old adage that information is best conveyed with “simple declarative sentences”. This would certainly help information be conveyed to all readers especially those in the 90% of the world where English is not peoples first language.
We thank the reviewer for this comment. We have rewritten some sentences of the manuscript to improve the clarity of the text.
Line 189. Meaning was not clear. Could you replace, “with respect to” with “were among”?
We changed the sentence, now at line 193, from “a lower percentage of current smokers with respect to those with negative results” to “a lower percentage of current smokers than among those with negative results”.
Line 189. The result that people who smoke are less likely to get COVID was surprising, because long term smoking can cause irreversible lung damage. Thus, this is a very important observation that should have been dealt with in the discussion. If the smoking result is to described in another paper, then the authors must say so. If the authors did not elaborate on this observation, because others have made it previously they should say so, and provide references.
We thank the reviewer for having highlighted this point. The association between smoking habit and SARS-CoV2 infection is still an open and debated issue. Smokers are at greater risk for respiratory infections however, emerging evidence suggests that the risk of SARS-CoV-2 infection may be lower among smokers compared to non-smokers but no definitive conclusion can be drawn (Grundy, Emily et al. "Smoking, SARS-CoV-2 and COVID-19: A review of reviews considering implications for public health policy and practice." Tobacco Induced Diseases, vol. 18, no. July, 2020, 58. doi:10.18332/tid/124788). This topic is highly relevant and will be specifically explored in a manuscript that is now in preparation. Following the reviewer comment, we have briefly commented our results in the discussion section (lines 321-325) and provided a reference ([36]).
Further consideration. The discussion of the association with the pneumococcal and flu vaccines with less covid-19 has been well discussed. There is an additional explanation that the authors may wish to include. If co-infections with flu or pneumococci result is a worse course of Covid-19 infection, then such individuals would be more likely to be tested and declared Covid-19 positive. In other words, by keeping flu and bacterial causes of pneumonia under control, the outcomes of co-infections with Covid-19 may be less severe on average.
We thank the reviewer for this insightful suggestion. We have added it in the Discussion the following sentences (lines 314-320): “Finally, recent studies suggested that influenza vaccination could mitigate the severity of COVID-19, either by modulating immune response or by reducing the risk of co-infections [16,17]. If confirmed, this issue should be considered in interpreting our results since non-vaccinated individuals may have presented more severe COVID-19 onset and course that made them more likely to undergo a NPS compared with vaccinated people. However, further studies are needed to verify these hypotheses and to explore the differences noted between younger and older adults.”

Reviewer 2 Report
The detail of recommendation is following:
Methodology
- Author should be added the detail of inclusion and exclusion criterion of your sample size.
- Author should be added the equation of sample size calculation. - Author should be added the detail of measurement pre-test.
- Author should be added the cut-point in each factors.
- Author should be added the map of your sample area.
Discussion
- Author should be updated the discussion part with current publication.
- Author should be added the recommendation from your finding.
Author Response
20 August 2020
Subject: Manuscript ID: vaccines-908602 - Minor Revisions (by 24th August 2020)
Dear Editor,
herewith we resubmit the revised version of our manuscript entitled “The association between influenza and pneumococcal vaccinations and SARS-CoV-2 infection: data from the EPICOVID19 web-based survey”.
We are grateful to the reviewers for their time and constructive comments. We have implemented their comments and suggestions and wish to submit a revised version of the manuscript for further consideration in the journal. Below, we provide a point-by-point response to each remark with a description and referral to all amendments made in the revised version of the manuscript. Changes in the manuscript are highlighted using track changes method.
We thank you and look forward to hearing from you.
Yours sincerely,
On behalf of the co-authors
Marianna Noale
Reviewer 2:
Methodology
- Author should be added the detail of inclusion and exclusion criterion of your sample size.
The study inclusion criteria were described at lines 103-105. We have now tried to make them clearer (lines 105-108): “The inclusion criteria for participation in the survey were: i) being 18 or older; ii) having access to a mobile phone, computer, or tablet with internet connectivity; iii) giving informed on-line consent to participate in the study. Those who did not meet these requirements were excluded from the study sample.”)
- Author should be added the equation of sample size calculation.
EPICOVID19 is based on a volunteer, self-selection survey sampling strategy. For this reason, no a priori sample size calculation was performed, but a convenience sample was considered. This aspect has been now clarified in the manuscript (lines 100-101): “EPICOVID19 is an Italian cross-sectional, web-based survey that was initiated in April 2020. A national convenience sample of volunteers was considered.”
- Author should be added the detail of measurement pre-test.
We are not sure to correctly understand the reviewer's observation; however, we confirm that EPICOVID19 is a cross-sectional study (therefore no pre- and post- test are available) and that no a priori sample size calculation has been made.
- Author should be added the cut-point in each factors.
We have now added cut-point for each variable considered in the multivariable models reported in Table 3 (lines 218-221): “*adjusted for sex (males vs females), age (years, continuous for “all model”; dichotomized into <67vs ≥67 for “≥65 years model”; dichotomized into ≥54 vs <54 for “<65 years model), education (secondary school diploma, university degree or post-graduate degree vs secondary school license, elementary school or less), Italian area of residence (area 1 vs (3,4); area 2 vs (3,4)), dichotomized self-reported diseases (CVD, hypertension, lung diseases, diabetes treated with medications, metabolic diseases), smoking status (current smoker vs no smoker or former smoker) and contact with confirmed COVID-19 cases (yes vs no).”
- Author should be added the map of your sample area.
A map with response rates for each Italian Region (x 10.000 inhabitants) has been now added (Supplementary Figure S1).
Discussion
- Author should be updated the discussion part with current publication.
We thank the Reviewer for this suggestion. We have now updated the references of our study in light of the most recent papers published on the topic (e.g., Marín‐Hernández D et al, J Med Virol, 2020 Jun 4; doi: 10.1002/jmv.26120.; Jehi L et al, PLoS One. 2020 Aug 11; doi: 10.1371/journal.pone.0237419.)
- Author should be added the recommendation from your finding.
We have now revised the study Conclusions to make clearer the possible implications of our findings (lines 349-354): “However, considering that multiple respiratory co-infections can frequently lead, especially in older age, to fatal respiratory failure, our findings support the need for collaborative public health programs to enhance the anti-pneumococcal and influenza vaccination campaign over the coming months. Special attention should be paid, in particular, on the most vulnerable categories of individuals who are at higher risk of negative COVID-19 prognosis.”
